# Analysis of the Extrusion Process of Aluminium Alloy Profiles

**DOI:** 10.3390/ma15238311

**Published:** 2022-11-23

**Authors:** Teresa Bajor, Anna Kawałek, Szymon Berski, Henryk Jurczak, Jacek Borowski

**Affiliations:** 1Faculty of Production Engineering and Materials Technology, Czestochowa University of Technology, 42-201 Czestochowa, Poland; 2Albatros Ltd., ul. Południowa 36, 78-600 Wałcz, Poland; 3Łukasiewicz Research Network—Poznań Institute of Technology, ul. Ewarysta Estkowskiego 6, 61-755 Poznań, Poland

**Keywords:** extrusion, aluminium alloys, car bodies, plastometric tests, rheological properties, FEM modelling

## Abstract

The paper presents an analysis of the results of numerical tests of the extrusion process of structural panels made of the 5xxx and 6xxx series aluminium alloys in a designed split die. The obtained products are intended for innovative superstructures of special car bodies. The main purpose of the research was the designed split die and numerical simulations and analysis of test results to determine the parameters of the extrusion process. The distribution of stress intensity, strain, strain rate, and temperature in the extruded metal was analysed for two different speeds of the punch movement. On the basis of the analysis of the distribution of stress values occurring in the extrusion process, the conditions enabling the real process of extrusion of the panel profile in industrial conditions in the designed split die were determined. It was shown that panel sections can be produced from ingots with a length of 770 mm on a press with a pressure of 35 MN (12”).

## 1. Introduction

Aluminium alloy flat bars (panels) with various, often complex, structures, which are usually made by extrusion on presses, are common products in modern, lightweight structures. Extrusion is a widely used process in the production of long components with complex cross-sections. This technological method allows for obtaining unique shapes of products, which provides exceptional design opportunities for architects and designers [1,2,3,4]. Extrusion is one of the types of metal forming processes. During the extrusion process, the material placed in the container with the die, subjected to the pressure of the punch, flows out through the holes of the die and is elongated at the expense of reducing the cross-section.

Aluminium alloys with magnesium of the 5xxx series are intended for plastic working. At the same time, these alloys are characterised by high strength properties, thanks to which they are very frequently used in various types of structures. These kinds of alloys are also characterised by very good technological properties, such as high corrosion resistance, high strength, good weldability, and ease of shaping. Moreover, they have the lowest density of all aluminium alloys, which reduces the weight of a given element, device, or entire structure. As a result, they are used in various economic sectors, and their percentage share in material processing is constantly growing. Currently, aluminium alloys are the most widely used in transport due to their low weight. At the same time, they are characterised by high strength properties and very good corrosion resistance, even in aggressive environments. In the automotive industry, aluminium alloys are used both for structural elements (frame and body elements), as well as engine parts and accessories. The use of aluminium frames significantly reduces the weight of the means of transport, which makes it easier to control, because its centre of gravity is lowered, and the braking distance of the vehicle is shortened [5,6,7].

6xxx series aluminium alloys contain magnesium and silicon. A characteristic feature of this series of alloys is very high corrosion resistance and good plasticity. These alloys are widely used in construction, interior design, the automotive industry, furniture, electronics, lighting, load-bearing elements of trucks, buses, ships, cranes, wagons, bridges, and barriers, as well as in the mining, chemical, food and shipbuilding industries [8,9,10,11].

Aluminium alloys have also been used as an effective method to improve the corrosion resistance and formability of Mg alloys. Magnesium and its alloys, such as aluminium alloys, are used in the automotive and aerospace industries due to their low density, good mechanical properties, excellent electromagnetic shielding ability, and good recyclability. However, the practical use of Mg alloys is limited by their low corrosion resistance [12]. Authors have presented effective methods of joining various Al/Mg alloys, with these methods also including the hot extrusion method [12]. A review of the literature showed that using the hot extrusion method can combine different materials: 1060/6063 Al alloys. The first reason is that the split metal streams are fresh without oxidation and impurities. Secondly, the materials inside the welding chamber experience high temperature, high pressure, and an almost vacuum atmosphere, which is beneficial for solid bonding and microstructure improvement [13].

The corrosion resistance of aluminium alloys can be increased by using the anodising process [14,15,16]. This process consists in creating a coating on the aluminium surface, which increases resistance to various external factors, such as acid rain, sea water, or UV radiation. It is also an effective way to delay the aging of aluminium. Anodising significantly improves the aesthetic value of aluminium products.

The use of extrusion technology of new large-size aluminium profiles with a width of 200 to 600 mm will allow the introduction to the market of a new solution in the form of an innovative superstructure for special-purpose vehicles, characterised by a modular body structure made of aluminium alloy panels. Thus far, no such solution exists in which extruded aluminium panels (especially from 5xxx and 6xxx series alloys) are used to build superstructures of special-purpose vehicles. Currently, the bodies of special vehicles are based on steel structures combined with composite components or steel sheets. Plastic bodies have elements joined in the gluing technology, while traditional bodies based on steel and aluminium are most often joined by welding or riveting. The use of an innovative modular superstructure with wide panels made of aluminium alloys, as proposed by the authors, eliminates the current problems, such as: very limited recycling possibilities of plastic elements, the harmfulness of substances used in the manufacturing process, and the length of the production cycle, as well as the nuisance associated with the processes of joining elements. Certainly, there are many solutions on the market that use aluminium profiles with different groove shapes and mounting methods, but there are no solutions using such wide profiles. Thanks to the use of panels, the assembly and disassembly time of bodies can be significantly reduced; individual panels can be easily replaced, which is also a very important factor in the operation of special-purpose vehicles that work in difficult conditions; their components are frequently damaged. The use of an aluminium modular superstructure made of large-size aluminium panels will allow the reduction in the weight of the vehicle by about 10% in relation to bodies currently made of welded structural steel. Moreover, aluminium is a material that helps to reduce the amount of greenhouse gases emitted into the environment. This is carried out in two ways: thanks to the use of bodies made of aluminium panels, the load capacity of the vehicles is increased, which, in turn, leads to an increase in transport capacity and the possibility of transporting more goods; the weight of the vehicles is further reduced, which helps reduce fuel consumption [1,17,18,19,20,21].

## 2. Purpose and Scope of the Study

The development of deformation technology at the design stage gives indisputable benefits when planning the production process of new products. The use of modern IT tools enables a multi-faceted analysis of changes occurring during the manufacturing process of the finished product. The purpose of the study was to design a die and to perform and analyse the results of numerical tests of the extrusion process of structural panels made of 5xxx and 6xxx series aluminium alloys intended for innovative construction of bodies of special-purpose vehicles. The influence of the following parameters of the extrusion process was analysed: distribution of strain intensity, distribution of temperature of the extruded alloy, distribution of strain rate, and distribution of stress intensity. The calculations were made using FEM for the three-dimensional state of deformation, taking the thermal phenomena occurring during the applied deformation scheme into account. Numerical modelling of the panel profile extrusion process was performed using the Forge^®^NxT software based on the finite element method [18,22,23,24,25,26]. The main focus of the FEM modelling process in Forge^®^NxT was the analysis of the distribution of the above-mentioned parameters during the extrusion of three Al alloys and the assessment of their suitability for the actual shaping process in a vertical hydraulic press with a container diameter of 12” and a pressure of up to 3500 T.

## 3. Materials and Methods

The material chosen for the tests was aluminium alloys with chemical compositions as given in Table 1.

Before the extrusion process modelling, plastic flow models for the three analysed alloys were developed: Al5754, Al6005, and Al6082. Plastic flow models were developed based on uniaxial compression tests performed with the Gleeble 3800 metallurgical process simulator. Plastometric tests were carried out for the following parameters:-Temperature: 350 °C, 400 °C, 450 °C, 500 °C, and 550 °C.-Deformation velocity: 0.01 s^−1^, 0.1 s^−1^, 1 s^−1^, 10 s^−1^, and 30 s^−1^.-Actual deformation: max. 1.15 for the GLEEBLE 3800 simulator.

The samples were heated at a constant rate of 5 °C/s to the desired temperature, held at this temperature for 20 s, and then deformed.

The results of plastometric tests were used to determine the coefficients of the yield stress function, which were used to simulate numerical extrusion processes in the Forge^®^NxT software. *σ_p_* formula (1) was used to determine the value of the yield stress. Table 2 presents the values of the parameters *A* and *m*_1_–*m*_9_ used to define the value of plasticising stress *σ_p_* for the three aluminium alloys [22].

The values of the parameters obtained as a result of the approximation of Equation (1).
(1)$$\sigma_{p}=Ae^{m_{1}T}\varepsilon^{m_{2}}{\dot{\varepsilon }}^{m_{3}}\varepsilon^{\frac{m_{4}}{\varepsilon }}{\left(1+\varepsilon \right)}^{m_{5}T}\varepsilon^{m_{7}T}{\dot{\varepsilon }}^{m_{8}T}T^{m_{9}}$$
where: $\sigma_{p}$—plasticising stress; $\dot{\varepsilon }$—strain rate; *T*—temperature of the deformed material; ε—actual deformation, A; *m*_1_–*m*_9_—function coefficients.

Equation (1) is frequently used to derive value *σ_p_* in computer software for numerical modelling of plastic working.

Modelling of the extrusion process was performed for each alloy with two different punch speeds. The main simulation parameters for all analysed alloys are presented in Table 3.

In order to carry out the extrusion process of the section shown in Figure 1, a die had to be designed, thanks to which it would be possible to obtain panels with a width of s = 200 mm and a thickness of h = 35 mm from an ingot homogenised in the casting process with 770 mm in length and 305 mm in diameter. The section consists of three parts: connector (I), central part (II), and outer part (III), and it is a component of load-bearing structures used in the automotive industry. To carry out the extrusion process, a multi-port die was designed with bridges separating the extruded material in such a way to obtain an even plastic flow of the metal in the calibration zone of the die.

In this type of die, during metal deformation, both in the real process and during process modelling using the finite element method, metal deformation occurs in several characteristic stages, as shown on the example of a port die with longitudinal and transverse bridges in Figure 2.

In the first stage of the extrusion process, the metal in the container contacts the die and flows into the ports, where it is separated by port bridges, both transverse and longitudinal (Figure 2a).

From the data presented in Figure 2a, a local increase in the metal flow velocity can be observed. In the next stage (Figure 2b), the metal fills the ports and flows into the welding chamber. At this stage, the value of the metal flow velocity decreases. In the last step of the extrusion process, when the welding chamber is filled with metal, the value of the plastic flow velocity increases again and reaches a maximum when passing through the calibration zone.

The metal flow kinematics in this type of die is determined by the distribution of the parameters that affect the plasticisation of the material, such as temperature, deformation, or strain rate. To obtain a section with the cross-section shown in Figure 1, a die was designed, the essential elements of which are shown in Figure 3.

Due to the small wall thickness of the panel section (g = 1.8 mm, and the largest is g_max_ = 3 mm) in relation to its width (s = 200 mm) and the symmetrical arrangement of the cutout, the analysis covered the distributions on the contact surface of the deformed metal with the die.

At the first stage of forming the section walls, deformation unevenness results from different metal flow velocities along the axis, which causes the formation of tongues (uneven shape of the beginning of the strand) and, therefore, the first stage of metal flowing out of the calibration part of the die was analysed.

The largest metal deformations during the extrusion process in this type of dies occur at two characteristic stages of the process. These are the stages in which the metal initially flows perpendicular to the axis of the ingot, and then, due to the shape of the die, the direction of its flow is forced to change to parallel to the axis of the ingot. Inner layers of the metal that are not in direct contact with the curved surface (edge) of the die have a slower flow velocity than those in direct contact with the surface of the tool. In these zones, there is a large velocity gradient, which causes areas of large deformation in the subsurface layers. Such a concentration of deformations in these zones is related to the tool geometry (edge rounding radius and edge inclination angle), which is not affected by the type of deformed material, the plastic resistance to metal flow resulting from the friction of external metal layers against the tool, and the internal friction depending on the material properties of the deformed metal.

During metal deformation in the designed die, there are two areas characterised by a high concentration of deformations. The first area is the zone where the metal flows from the container to the ports during metal separation. The second is a two-stage area of metal outflow from the welding chamber to the calibration part, where the panel profile is given the final shape. Modelling of the extrusion process of the panel section shown in Figure 1 was performed for three different 5xxx and 6xxx series Al alloys. Moreover, in order to adapt the modelled process to the real conditions of one of the industrial aluminium presses, the process was carried out for two punch speeds: 3 and 6 mm/s.

## 4. Analysis of Modelling Results of the Panel Profile Extrusion Process Using the Forge^®^NxT Software

### 4.1. Analysis of Strain Intensity Distributions

Figure 4, Figure 5 and Figure 6 show the distribution of strain intensity during the process of extrusion of a panel section from three different Al alloys.

For all tested materials and variants of the extrusion process, two areas with a high concentration of strain intensity can be observed. The differences result from the size of these areas and the value of deformations.

For 5754 aluminium at the punch speed *v* = 3 mm/s (Figure 4a) in the first area, the highest values of strain intensity are observed (ε = 10), mainly in the central metal zones of the base of the port walls (port entrances) and in small fragments in the contact area of the bridge surfaces with the port walls. By contrast, for the maximum values of deformation on the same surfaces, but for the 4 ports furthest from the axis of the ingot, the values of strain intensity are, on average, between 8 and 9. In the second area, the highest strain intensity is observed at the edge of the second stage of cross-sectional reduction, in the places where the section wall is thickest, and in the corners of individual parts of the section. One can also observe a large unevenness in strain intensity in parts I and II, their value ranging from 5 to 9. The lowest values of strain intensity occur in the welding chamber, in the corners of the ports, where the metal has the lowest flow velocity and specific dead zones are formed. The same phenomenon occurs in the corners of the container, where the values of strain intensity are also very small.

Increasing the extrusion speed to 6 mm/s (Figure 4b) causes an increase in the zones with maximum values of strain intensity in the first area, and for the ports that are furthest from the axis of the ingot, strain intensity increases to the value of 10. However, in the second area, due to higher values of the metal flow velocity and the uniform flow of the metal along the axis of the ingot, the values of strain intensity range from 4 to 6.

For 6005 aluminium extruded with a punch speed of *v* = 3 mm/s (Figure 5a), the highest values of strain intensity equal to 10 were observed in the first area, mainly on the entire contact surface of the bridges with the metal and at the base of the walls of all ports. In the second area, the highest strain intensity is observed in the same places as for the Al 5754 alloy. However, the unevenness of the distribution of strain intensity is much greater, because the area with the greatest deformations is larger and in part III, it covers almost 50% of its length, while in parts I and II, the values of strain intensity vary in the range from 5 to 10. The lowest strain intensity occurs in the welding chamber in the corners of the ports and in the corners of the container, and its value is from 0 to 1.

For this Al alloy, an increase in the extrusion speed to 6 mm/s (Figure 5b) results in a more even distribution of the strain intensity in the second reduction stage, where the non-uniformity between strains is in range from 2 to 5, while in the first reduction stage, these irregularities are significant and in the range from 4 to 10. In all three parts (I–III), the maximum values of the strain intensity appear in the places where the profile walls are thickest, because in these places, the metal has a higher flow velocity than in the neighbouring thin-walled zones. This difference in flow velocity causes large deformations in these places.

During the extrusion process of the 6082 aluminium alloy with the punch speed *v* = 3 mm/s (Figure 6a), it can be observed that in the first area, the zone of maximum deformation values is very extensive and virtually covers the entire area of the base of metal entry into the ports and reaches up to 20% of the height of the ports. In the second area, deformation unevenness is very high, and deformation values range from 2 to 10.

By increasing the extrusion speed of this alloy to 6 mm/s (Figure 6b), the area containing the deformation with maximum values extends to about 25% of the port length in the first area, and in the second area, the distribution of deformation in individual parts is almost the same as the distribution obtained for the punch speed of 3 mm/s.

On the basis of the tests carried out, it is found that the increase in the extrusion speed increases the strain intensity in specific zones. The distributions of strain intensity values obtained on the cross-section of the extruded profile depend on the thickness of the profile wall.

### 4.2. Analysis of Temperature Distribution in the Extruded Metal

The temperature distribution in the process of extrusion of sections depends strictly on the temperature conditions of the tools and their shape, as well as on the process conditions—in this case, extrusion speed and friction. Moreover, the material properties of the extruded metal are extremely important, as they significantly affect the plastic flow resistance of the metal during extrusion and have a significant share in the value of energy and strength parameters of the process. There are only few plants in Poland and in the world where it is possible to carry out the process of extrusion of this type of section, due to the need to use containers with large diameters and complicated shapes of dies, which results in a large energy expenditure, possible only in presses with very high pressures. Therefore, in the next stage of the study, an analysis of the temperature distribution during the process of extrusion of a panel section from three different Al alloys was carried out (Figure 7, Figure 8 and Figure 9). Simulations of the extrusion process were carried out for two different punch speeds: 3 and 6 mm/s. The analysis of the temperature distribution was performed mainly in the calibration zone, i.e., in the area where the plastic metal flows out from the ports in the welding chamber zone to the calibrating strips.

For the Al5754 alloy and the extrusion speed *v* = 3 mm/s (Figure 7a), the temperature values range from 485 °C to 492 °C. Higher temperature is observed in parts I and III, where the section has a thickened wall, where there is a less intense heat exchange with the surroundings, and where large deformations occur. Increasing the punch feed speed to 6 mm/s (Figure 7b) affects the unevenness of the temperature distribution in the individual parts, as well as in the entire volume of the die. The same temperature of approximately 492 °C occurs along the entire calibration strip in part II, while in part III, the temperature ranges from 492 °C to 499 °C. In part II, a temperature distribution from 488 °C on the thickened wall to 499 °C on the free ends of the section can be observed. For this material (Figure 7b), the temperature in the corners of the ports in the welding zone drops to 480 °C.

For aluminium 6005 extruded with a punch speed of *v* = 3 mm/s (Figure 8a), the temperature distribution is uniform for all parts and ranges from 478 °C to 485 °C. After increasing the punch feed speed to *v* = 6 mm/s (Figure 8b), the metal slightly cools down at the ends and reaches temperatures in the range of 478–485 °C, in relation to the temperature of the metal along the axis of the ingot in part II, where it reaches the temperature of 492 °C.

When extruding aluminium 6082 at a lower punch feed speed (Figure 9a), an even temperature is observed in the entire volume of the metal contained in the die in the range from 478 °C to 484 °C. Increasing the extrusion speed (Figure 9b) does not change the nature of its decomposition, but causes an increase by about 7 °C.

### 4.3. Analysis of the Distribution of Strain Rate

Modelling the extrusion process of a panel section in a die with a constant shape causes the metal in all variants of the process to have the same plastic deformation path, which means that extrusion speed (*v*) has the greatest impact on the distribution of strain rate values. The temperature of the deformed metal and its rheological properties also have a significant effect on the distribution of strain rate. Process analysis with the use of FEM enables assessing the sensitivity of the material to strain rate.

Numerical analysis of strain rate for the panel section extrusion process allowed for determining the value ranges of this parameter. For all analysed Al alloys and two tested punch feed rates, the intensity of the extrusion speed is in the range from 0 to 3 s^−1^. This is the lower limit of the value of strain rate that occurs during the extrusion process. In the literature [27,28], it is assumed that for hydraulic presses, it is in the range from 3 to 10 s^−1^ and in mechanical presses, from 20–80 s^−1^.

For alloy Al5754, the distribution of strain rate is shown in Figure 10a. Disregarding the values for erroneous mesh nodes (red—escape of nodes from the port), intensity strain and velocity are in the range from 0 to 1.5 s^−1^. The greatest strain rate is observed in the welding zone, where the metal filling the die changes its direction rapidly and the value of the plastic flow rate changes as well. The change in the flow direction is forced by the tool geometry and is limited in this area: by the die core, the edges of the ports, and the bridges (areas marked with a solid line). In this zone, the highest values of strain rate range from 0.3 to 1.2 s^−1^, whereas in the zone of two-stage reduction of the cross-section of the metal flowing from the welding chamber to the calibration part (area marked with a dashed-dotted line), the values of strain rate range from 0.3 to 1.5 s^−1^.

When the extrusion speed is increased to 6 mm/s (Figure 10b), areas with higher values of strain rate are greater, and the maximum values of the average strain rate reach up to 2.1 s^−1^.

In the process of extrusion of alloy Al 6005 deformed at a speed of *v* = 3 mm/s (Figure 11a), in the welding zones (marked in Figure 10a with a solid line), the distribution of strain rate is similar to the distribution obtained for alloy Al5754. There are areas of increased intensity of strain rate in the shape of cones, where the average values range from 0.3 to 0.9 s^−1^; however, the distribution of strain rate is more uniform. An uneven distribution of strain rate can be noticed in the area of the two-stage reduction in the cross-section, where the values of strain rate are up to 2.1 s^−1^.

Increasing the extrusion speed to *v* = 6 mm/s (Figure 11b) increases the maximum values of strain rate, which is about 3 s^−1^. The average values of this parameter in the zone where the material exits from ports (for the area marked with a solid line in Figure 10a) are slightly higher than for the extrusion process with punch feed speed *v* = 3 mm/s and range from 0.3 to 1.2 s^-1^. Similarly, comparing the metal flow during the extrusion process with velocity *v* = 3 mm/s, the area with an uneven distribution of strain rate is slightly larger in the zone of two-stage reduction of the cross-section (part II), and the value of strain rate increases to 2.4 s^−1^. Zones with the highest values of strain rate occur at the height of the cones of intense flow resulting from the zone where the metal passes from the port bridge to the calibration zone.

In the process of extrusion of alloy Al 6082, the areas with the highest values of strain rate occur when the aluminium flows into the port channels in the material separation zone, and the values of this parameter reach 3 s^−1^. In part II also, in the first and second reduction stage, in the central part of the blank, the maximum values of strain rate appear (Figure 12a).

By increasing the extrusion speed, it is possible to obtain the effect of a more even distribution of strain rate over the entire cross-section of the section, which is particularly visible in part I in the first reduction stage (Figure 12b).

### 4.4. Analysis of the Distribution of Stress Intensity

Figure 13, Figure 14 and Figure 15 show the distribution of stress intensity for the three tested Al alloys and for two punch feed speeds during extrusion of a panel section.

Figure 13a,b show the distribution of stress intensity during the alloy Al 5754 extrusion process deformed at two different punch feed speeds. From the data provided in these figures, it can be observed that for this Al alloy, the largest areas of maximum stress intensity occur in the places where the aluminium flows into the ports and on the edges separating the ports, whereas in the calibration zone, the maximum values of stress intensity occur only in single nodes of the second reduction stage.

In Figure 13b, showing the distribution of stress intensity during the extrusion process at a speed of 6 mm/s, it can be observed that increasing the extrusion speed results in a more uniform flow of the aluminium layers in the separation zone. It also includes small areas where the maximum values of stress intensity occur. By contrast, in the calibration zone, there is a slight increase in the value of stress intensity, both in the first and in the second reduction stage. It is especially visible in the axial zone (part II of the section—Figure 1), where the values of stress intensity increase from 20 to 60 MPa.

For aluminium 6005, the values of stress intensity in the same stage of the extrusion process in all characteristic areas are significantly lower than for alloy Al 5754. In the process of extruding the section with punch feed speed *v* = 3 (Figure 14a), the highest values of stress intensity occur in the calibration area (part II). It is especially visible in places where the distribution of strain rate (Figure 11a,b) includes cones of intense flow, and the values of stress intensity reach up to 60 MPa. In the reduction area in the first and second stage, there are different values of stress intensity from 20 MPa to 60 MPa. In the separation area (I), stress intensity is much lower and reaches a maximum of 40 MPa. By increasing the extrusion speed to 6 mm/s (Figure 14b) in area I (separation), the values of stress intensity slightly increase to about 50 MPa, whereas in part II, both in the first and second stage of reduction, the values of stress intensity are distributed more evenly than for the punch feed speed *v* = 3 mm/s and are in the range from 30 to 50 MPa. Apart from a greater uniformity of stress intensity, this alloy is not very sensitive to changes in strain rate.

Data in Figure 15 suggest that during the deformation of alloy Al 6082, the values of stress intensity both in the separation and calibration zones, as well as in the remaining area, are in the range from 10 to 30 MPa and slightly differ for both extrusion speeds tested.

Table 4 shows the values of punch pressure on the metal determined in the modelling process during the profile extrusion for all analysed alloys and the two punch feed speeds.

The data in Table 4 show that the highest values of the total pressure of the punch during the profile extrusion process occur for aluminium alloy 5754, and the lowest for 6082 alloy. Based on the presented results of numerical tests of the profile extrusion process, it can be concluded that it is possible to carry out the extrusion process of this product on a press with a pressure of 3500 T.

## 5. Final Statements and Conclusions

Based on the theoretical analysis of the study results of the extrusion process of the Al alloy panel section, the following conclusions were formulated:-Based on the obtained results, it is possible to define the conditions enabling the actual process to be carried out in an industrial plant in a designed split die.-Taking the actual rheological properties of the analysed Al alloys during numerical modelling of the extrusion process into account will ensure an increase in the accuracy of calculations in relation to the actual technological processes.-For all tested materials and variants of the extrusion process, two areas of high concentration of strain intensity can be observed. There are differences in the sizes of these areas and in the values of deformation.-The conducted numerical tests show that the temperature increase in the deformed material is related to plastic deformation and the friction phenomenon occurring between the material and the die.-The value of the pressing force during the process depends on the stage of the extrusion process. The first maximum local pressure force occurs when the material is separated by the port bridges, both transverse and longitudinal. In the next step, the metal fills the ports and flows into the welding chamber. At this stage, the value of the extrusion force drops slightly. As the sealing chamber is filled, the value of the extrusion force increases again and reaches its maximum when passing through the calibration zone.-Based on the analysis of the distribution of values of force occurring in the extrusion process, it can be concluded that panel sections can be produced from ingots with a length of 770 mm using a press with a pressure of 35 MN (12”), because the maximum extrusion force does not exceed 30 MN.

## Figures and Tables

**Figure 1 materials-15-08311-f001:**
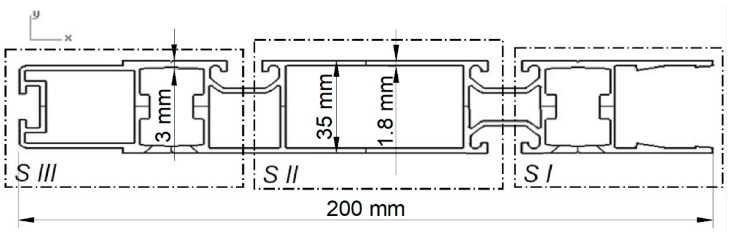
Panel section of an arm (component) for load-bearing structures used in the automotive industry with dimensions of 200 mm × 35 mm.

**Figure 2 materials-15-08311-f002:**
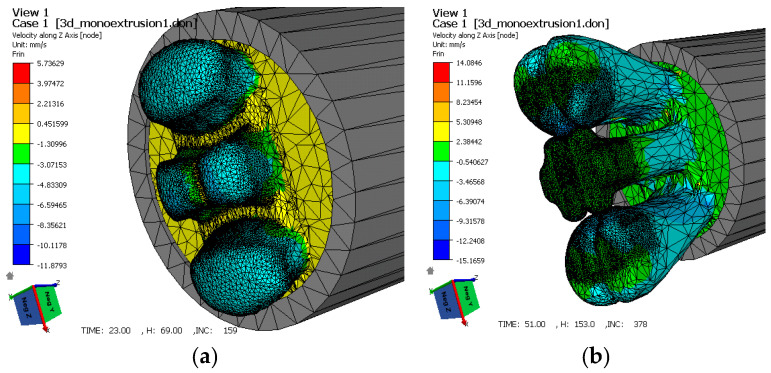
Stages of metal deformation in port dies with separating bridges (edges), a core, and a welding chamber: (**a**) metal is separated in the core part of the die through port bridges; (**b**) metal flows into the welding and calibration zone through contact with the plate part of the die.

**Figure 3 materials-15-08311-f003:**
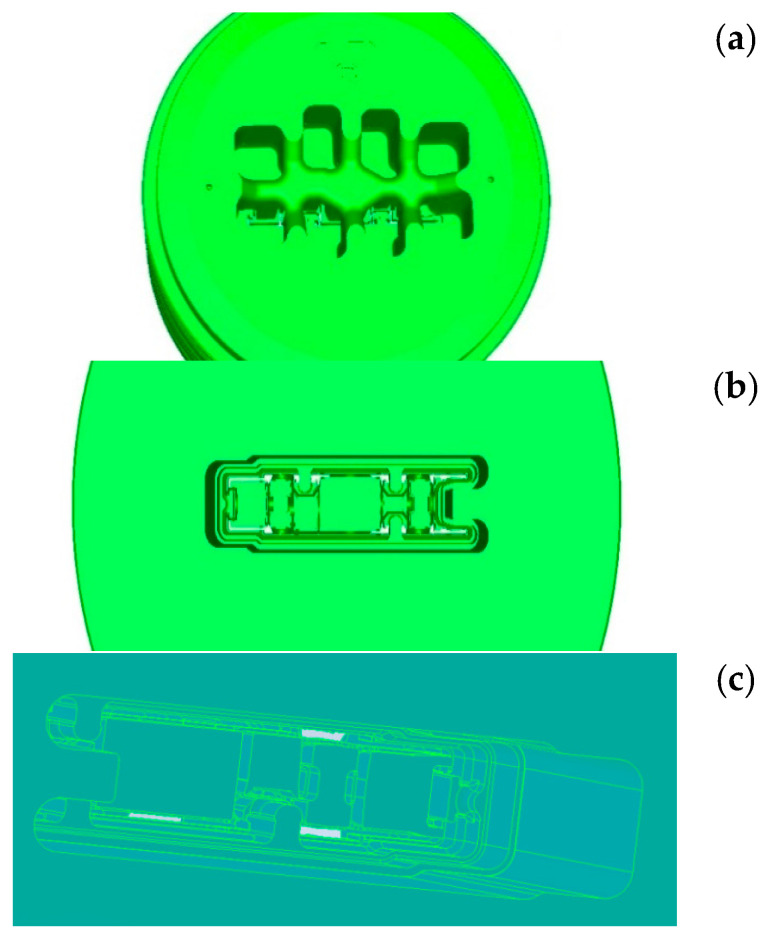
Die structure: (**a**) inlet part to the die with port opening; (**b**) two-stage calibration part of the die; (**c**) outlet part of the die, where the metal temperature stabilises after leaving the calibration zone (not used in the first modelling stage).

**Figure 4 materials-15-08311-f004:**
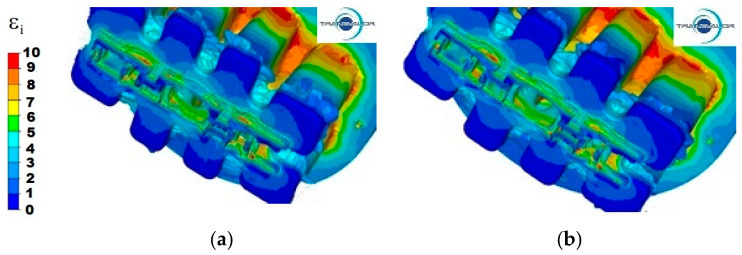
Distribution of strain intensity—Al 5754 alloy: (**a**) punch feed speed 3 mm/s; (**b**) punch feed speed 6 mm/s.

**Figure 5 materials-15-08311-f005:**
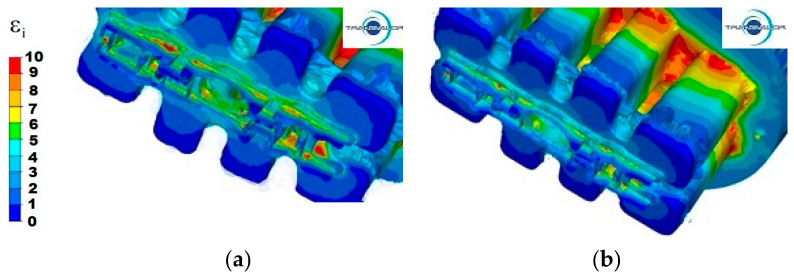
Distribution of strain intensity—Al 6005 alloy: (**a**) punch feed speed 3 mm/s; (**b**) punch feed speed 6 mm/s.

**Figure 6 materials-15-08311-f006:**
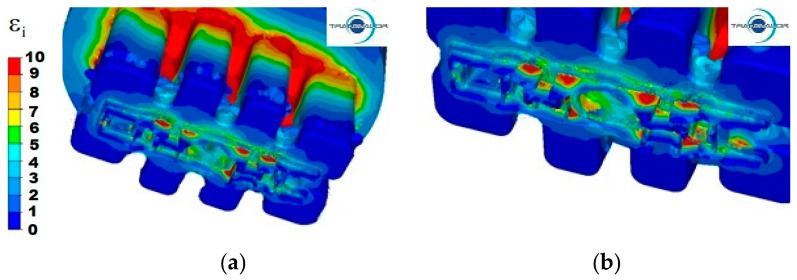
Distribution of strain intensity—Al 6082 alloy: (**a**) punch feed speed 3 mm/s; (**b**) punch feed speed 6 mm/s.

**Figure 7 materials-15-08311-f007:**
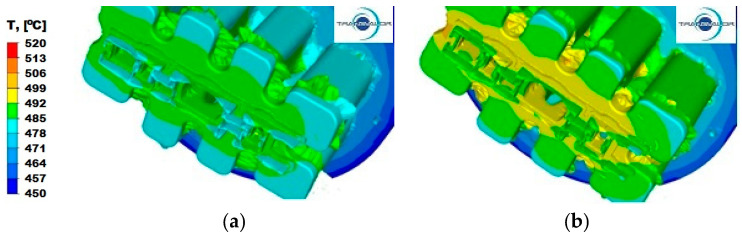
Temperature distribution—Al 5754 alloy: (**a**) punch feed speed 3 mm/s; (**b**) punch feed speed 6 mm/s.

**Figure 8 materials-15-08311-f008:**
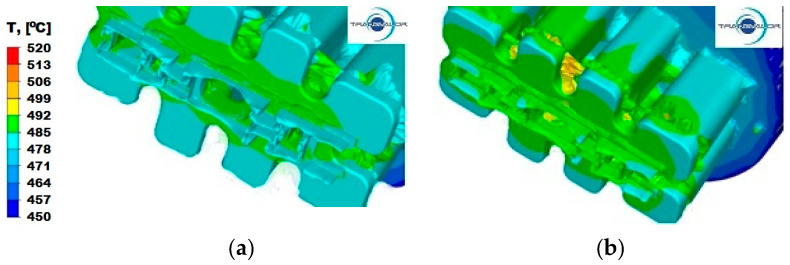
Temperature distribution—Al 6005 alloy: (**a**) punch feed speed 3 mm/s; (**b**) punch feed speed 6 mm/s.

**Figure 9 materials-15-08311-f009:**
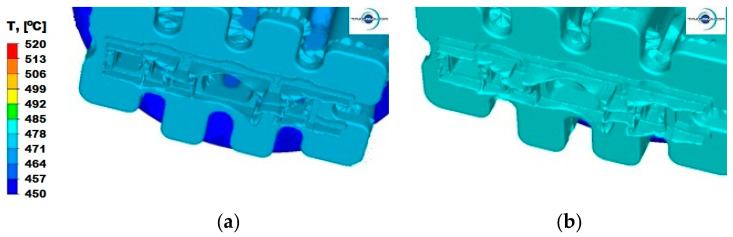
Temperature distribution—Al 6082 alloy: (**a**) punch feed speed 3 mm/s; (**b**) punch feed speed 6 mm/s.

**Figure 10 materials-15-08311-f010:**
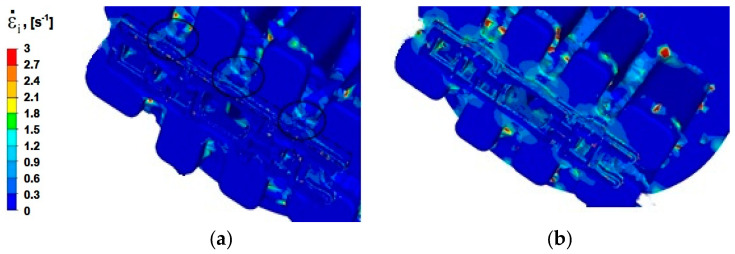
The distribution of strain rate—alloy Al 5754: (**a**) punch feed speed 3 mm/s; (**b**) punch feed speed 6 mm/s.

**Figure 11 materials-15-08311-f011:**
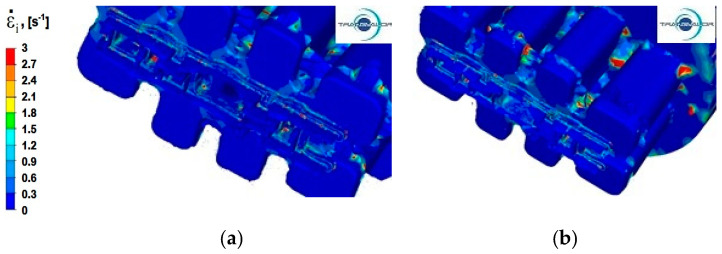
The distribution of strain rate—alloy Al 6005: (**a**) punch feed speed 3 mm/s; (**b**) punch feed speed 6 mm/s.

**Figure 12 materials-15-08311-f012:**
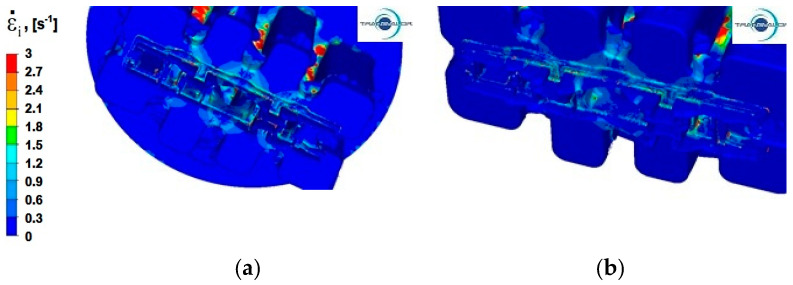
The distribution of strain rate—alloy Al 6082: (**a**) punch feed speed 3 mm/s; (**b**) punch feed speed 6 mm/s.

**Figure 13 materials-15-08311-f013:**
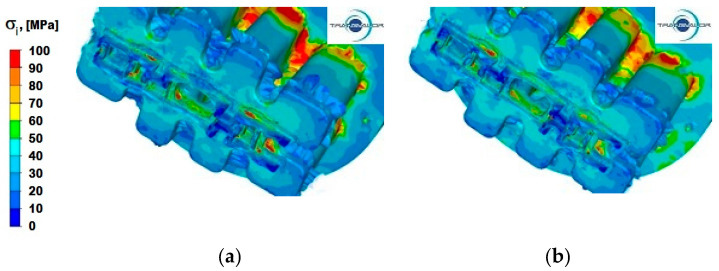
Distribution of stress intensity—alloy Al 5754: (**a**) punch feed speed 3 mm/s; (**b**) punch feed speed 6 mm/s.

**Figure 14 materials-15-08311-f014:**
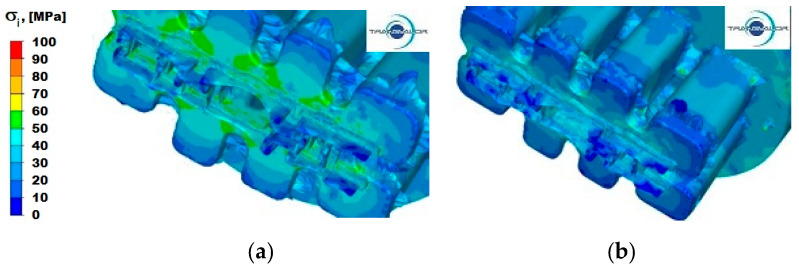
Distribution of stress intensity—alloy Al 6005: (**a**) punch feed speed 3 mm/s; (**b**) punch feed speed 6 mm/s.

**Figure 15 materials-15-08311-f015:**
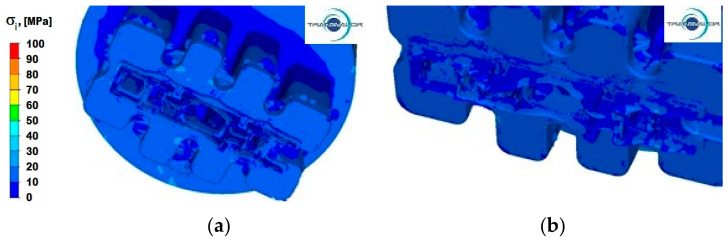
Distribution of stress intensity—alloy Al 6082: (**a**) punch feed speed 3 mm/s; (**b**) punch feed speed 6 mm/s.

**Table 1 materials-15-08311-t001:** Chemical composition of the investigated alloys [%].

Alloy	Si	Fe	Cu	Mn	Mg	Cr	Zn	Ti	Al
**5074**	0.224	0.14	0.007	0.465	3.44	0.002	0.002	0.018	R
**6005**	0.40	0.175	0.05	0.05	0.675	0.05	0.05	0.05	R
**6082**	0.95	≤0.18	≤0.02	0.50	0.95	≤0.03	≤0.02	≤0.02	R

**Table 2 materials-15-08311-t002:** Parameter values *A* and *m*_1_÷*m*_9_ used to define the *σ_p_* value of Al alloys.

Al Alloys		Al5754	Al6005	Al6082
The values of the parameters obtained as a result of the approximation of Equation (1)	** *A* **	0.1900358	79.928099	9.561 × 10^−7^
	** *m* _1_ **	−0.0074103	−0.0055896	−0.012197
	** *m* _2_ **	0.3359757	0.3994022	0.1363548
	** *m* _3_ **	−0.1777271	−0.0724108	0.1500382
	** *m* _4_ **	−0.0002228	−1.323 × 10^−5^	−0.000265
	** *m* _5_ **	−0.0042127	−0.0013238	−0.0005808
	** *m* _7_ **	0.4302946	−0.4114212	0.0370012
	** *m* _8_ **	0.0007222	0.00043599	−2.658396
	** *m* _9_ **	1.6723026	0.42435319	3.8141389

**Table 3 materials-15-08311-t003:** Main parameters used in extrusion process for all Al alloys.

Extrusion Speed v, mm/s	Initial Temperature of Billet T, °C	Friction Coefficient between Die and Ingot, μ	Friction Coefficient between Punch and Ingot, μ	Heat Transfer Coefficient α, kg/(°C·s^−3^)	Extrusion Ratio, λ
3	485	0.4	0.07	10,000	48
6					

**Table 4 materials-15-08311-t004:** Summary of values of the total punch pressure during profile extrusion.

Aluminium Alloy	The Speed of the Punch [mm/s]	Total Pressure [T]
**5754**	3	2500
	6	2800
**6005**	3	1200
	6	1300
**6082**	3	1100
	6	1300

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
