# Peer review of "Analysis of the Extrusion Process of Aluminium Alloy Profiles"

_materials, 2022, doi:10.3390/ma15238311_

Round 1
Reviewer 1 Report
In this paper, plenty of simulations were carried out on 5xxx and 6xxx series aluminum alloys. The simulation results under different extrusion speeds, such as stress intensity, strain, strain rate and temperature, were systematically and analyzed. The manuscript can be accepted for publication after revisions according to the following comments:
1. Co-extrusion is mentioned many times in this paper, while an introduction on co-extrusion is lacking. The authors should provide a detailed description of co-extrusion in the Introduction part and cite the relevant literatures. Several typical papers on co-extrusion are as follows:
- https://doi.org/10.1016/j.jmatprotec.2018.03.027
- https://doi.org/10.1016/j.jmatprotec.2017.07.009
2. The main dimensions in Figure 1 should be marked out.
3. How was the billet obtained? Was the billet homogenized before hot compression or hot extrusion?
4. The main simulation parameters such as extrusion temperature, extrusion speed, extrusion ratio, friction, transfer coefficient, etc. should be listed in the same table for comparison.
5. There are 28 figures in the paper, and some of the images are not very clear. It is recommended to provide clear images and merge the related images to reduce the number of figures.
6. The conclusion is too long and tedious.
Author Response
Dear Editor and Reviewers,
We appreciate your valuable comments and detailed suggestions on our manuscript “Analysis of the extrusion process of aluminium alloy profiles”, which was submitted to MATERIALS Special Issue
The revisions have been clearly highlighted using the "Track Changes" and “Comments” functions in Microsoft Word, therefore they are easily visible to editors and reviewers. The detailed explanations of all comments of the reviewers are provided underneath. We hope the revised manuscript can meet the requirements and high scientific standard of MATERIALS.

Reviewer 2 Report
Dear Authors;
The work is very interesting, which focuses on the analysis of the extrusion process of aluminum alloy profiles using FEM modelling. The manuscript is well written and well organized. The work is original but not innovative. The scientific part of this work is quite interesting and up to the standard of the journal of “materials”.
I suggest that the authors should pay much attention to revise the manuscript by taking following points into account:
- The authors should more clearly emphasis the novelty of their work in the abstract and introduction.
- Abstract must include some explicit information about the results obtained rather than giving general statements.
- Introduction: I strongly suggest that the Authors add new references.
- Where is the comparison of those materials with the commercial products and other types of materials? A reader should understand the overall benefits of the studied materials.
- A comparison with the existing published literature and discussion of the results are missing.
- For all the figures: the values of effective strain are not clear. Authors should improve the resolution of the figures.
- In the section 4.1, authors should explain the phenomena.
- Interpretation of Fig. 8 (L263-L265): the results are not well interpreted. This section should be detailed.
- Each figure should be placed under the corresponding paragraph.
- L485-L487: “The data in Table 3 show that the highest values of the total pressure of the 485 punch during the profile extrusion process occur for the aluminum alloy 5754, 486 and the lowest for the 6082 alloy”. It should be explained.
Sincerely Yours,
Dr. Mohsen Mhadhbi
Author Response

(The authors gave the same response as above.)

Round 2
Reviewer 2 Report
Dear Authors;
Thanks for your work in revising your manuscript according to the indicated comments. The revised paper is well improved.
I hope that this revised paper can be acceptable for publication in the
journal of "Materials".
Good Luck!
Sincerely Yours,
Reviewer